# Rheological and Pipe Flow Properties of Chocolate Masses at Different Temperatures

**DOI:** 10.3390/foods10112519

**Published:** 2021-10-20

**Authors:** Vojtěch Kumbár, Veronika Kouřilová, Renáta Dufková, Jiří Votava, Luděk Hřivna

**Affiliations:** 1Department of Technology and Automobile Transport (Section Physics), Faculty of AgriSciences, Mendel University in Brno, Zemědělská 1, 613 00 Brno, Czech Republic; jiri.votava@mendelu.cz; 2Department of Food Technology, Faculty of AgriSciences, Mendel University in Brno, Zemědělská 1, 613 00 Brno, Czech Republic; veronika.kourilova@mendelu.cz (V.K.); renata.dufkova@mendelu.cz (R.D.); hrivna@mendelu.cz (L.H.)

**Keywords:** chocolate masses, temperature, rheology, thixotropy, pipe flow velocity, mathematical models, 2D and 3D flow profiles

## Abstract

Chocolate masses are one of the basic raw materials for the production of confectionery. Knowledge of their rheological and flow behaviour at different temperatures is absolutely necessary for the selection of a suitable technological process in their production and subsequent processing. In this article, the rheological properties (the effect of the shear strain rate on the shear stress or viscosity) of five different chocolate masses were determined—extra dark chocolate (EDC), dark chocolate (DC), milk chocolate (MC), white chocolate (WC), and ruby chocolate (RC). These chocolate masses showed thixotropic and plastic behaviour in the selected range of shear rates from 1 to 500 s^−1^ and at the specified temperatures of 36, 38, 40, 42, and 44 °C. The degree of thixotropic behaviour was evaluated by the size of the hysteresis area, and flow curves were constructed using the Bingham, Herschel–Bulkley and Casson models with respect to the plastic behaviour of the chocolate masses. According to the values of the coefficients of determination R^2^ and the sum of the squared estimate of errors (SSE), the models were chosen appropriately. The most suitable models are the Herschel–Bulkley and Casson models, which also model the shear thinning property of the liquids (pseudoplastic with a yield stress value). Using the coefficients of the rheological models and modified equations for the flow velocity of technical and biological fluids in standard piping, the 2D and 3D velocity profiles of the chocolate masses were further successfully modelled. The obtained values of coefficients and models can be used in conventional technical practice in the design of technological equipment structures and in current trends in the food industry, such as 3D food printing.

## 1. Introduction

Chocolate is generally very popular worldwide and consumed across all generations. Its popularity is closely related to its overall sensory properties [1]. In order for chocolate to be called chocolate, it must meet legislative requirements for each type of chocolate. Dark chocolate contains at least 18% by weight of cocoa butter, 14% fat-free cocoa components, and the total content of cocoa components should be at least 35%. Milk chocolate means a product containing at least 25% of total cocoa components, at least 14% of milk solids, at least 2.5% of fat-free cocoa components, at least 3.5% of milk fat, and at least 25% of total fat, which is the sum of the cocoa butter and milk fat [2]. White chocolate is made from cocoa butter, milk or dairy products, sweeteners, or other ingredients [3,4]. According to the current legislation, white chocolate must contain at least 20% cocoa butter, at least 14% milk solids, with at least 3.5% milk fat [2]. Recipe choice of dairy ingredient affects the strength of the finished product [5]. In contrast to the above-mentioned types of chocolates, there is also ruby chocolate. The Belgian-Swiss company Barry Callebaut brought ruby chocolate to the world market in 2017. The exact technological process of production is not known, it is the know-how of the above-mentioned company, which patented ruby chocolate in 2012 [6,7]. However, the standard for chocolate and chocolate products has not yet officially recognised ruby chocolate as the fourth type of chocolate [8]. Although this type of chocolate is not yet covered by European legislation, it can be called chocolate because, according to the information on the packaging, it contains at least 47% cocoa components and at least 26% milk solids. The basic component of quality chocolate is cocoa butter, in which the other components are dispersed, which differ according to the type of chocolate. Cocoa butter forms a so-called continuous phase in chocolate, which forms a network structure for the spatial retention of the dispersed components. Cocoa butter can crystallise in a number of polymorphic forms as a function of triglyceride composition, with fatty acid composition influencing how liquid fat solidifies. Cocoa butter has six polymorphic forms (I–VI), the principals being α, β, and β′. Form V, a β polymorph, is the most desirable form (in general) in well-tempered chocolate, giving a glossy appearance, good snap, contraction, and resistance to bloom. If chocolate is poorly tempered, the outcome is the β Form IV, which rapidly transforms into Form V. This influences colour as reflected light is disoriented by unstable, disorganised crystal growth. Untempered chocolate is soft and not effectively demoulded. In cocoa butter, Forms V and VI are the most stable forms. Form VI is difficult to generate although it is formed by lengthy storage of tempered chocolate accompanied by fat bloom. In addition, Form VI has a high melting temperature (36 °C) and crystals that are large and gritty on the tongue. The unstable Form I has a melting point of 17 °C and is rapidly converted into Form II that transforms more slowly into III and IV. Tempering has four key steps: melting to completion (at 50 °C), cooling to point of crystallisation (at 32 °C), crystallisation (at 27 °C), and conversion of any unstable crystals (at 29–31 °C). For chocolate to be in an appropriate polymorphic form, tempering is crucial, influencing final quality characteristics such as colour, hardness, handling, finish, and shelf-life characteristics [3]. Thus, liquid chocolate is a complex polydisperse system and the relationships between the individual components determine the physical and rheological properties of the chocolate [4,9,10]. As the shear rate increases, the chocolate becomes more fluid, i.e., has a lower viscosity, due to the disintegration of the polydisperse chocolate system [3,11,12]. The rheological and textural properties of the chocolate are affected by its specific composition, especially the content of the fats, sugars, milk, other milk components, and surfactants. The fat content of chocolates is usually in the range of 25–35% [3]. The fat content has a significant effect on changes in the plastic viscosity. If chocolate contains a 28–32% fat content, these changes are more pronounced than for chocolates with a fat content above 32% [12]. Melted chocolate exhibits non-Newtonian plastic fluid behaviour, with shear thinning when the yield stress is exceeded [3]. The magnitude of this shear stress depends on the temperature of the chocolate, in this case, an indirect ratio applies, so the more the chocolate is heated, the less shear stress is needed to initiate the flow, the so-called yield stress [13]. Rheological models according to Herschel–Bulkley and Casson are widely used to model flow curves, respectively, to model the behaviour of non-ideal plastic liquids, so they could be useful for chocolate materials.

Based on the introductory literature review, the following two scientific hypotheses were established. These hypotheses are not completely verifiable in the contemporary literature, so they are given due attention in this article.

Chocolate masses have a plastic property with shear thinning behaviour (pseudoplastic with a yield stress).With an increasing temperature, the plastic viscosity or the consistency (depends on used rheological model) and thixotropy of the chocolate masses decreases.

## 2. Materials and Methods

### 2.1. Samples of Chocolate Masses

This research includes and describes the behaviour of chocolate masses that do not contain cocoa butter substitutes. Five chocolate masses were used for the experiments—extra dark chocolate (EDC), dark chocolate (DC), milk chocolate (MC), white chocolate (WC), and ruby chocolate (RC).

The EDC (Domori, Italy) contains 100% cocoa components (Morogoro, Tanzania). The DC, MC, and WC (Belcolade, Switzerland) contain 55.5%, 36.5%, and 28% cocoa components. The RC (Callebaut, Belgium) contains 47.3% cocoa components. Further characteristics (nutritional data) of the chocolate masses are given in Table 1.

The individual chocolate masses were tempered (by the manufacturer) before distribution and used in this research in the form of chocolate drops. For completeness of sample characteristics, the specific tempering curves of chocolate masses are given in Table 2.

The chocolate masses were approximately the same age (difference of 2 weeks in the maximum between the productions of individual masses). Prior to analysis, the masses were stored under the same conditions in a single warehouse at 12 °C.

### 2.2. Rheological Measurements

The rheological properties of the chocolate masses were measured using an RST CC rotary rheometer (AMATEK Brookfield, Middleboro, MA, USA) with a cone-plate measuring geometry. The diameter of the cone was 50 mm with an angle of 2° (the standard designation is RCT-50-2). The gap between the cone and the plate was set to 0.05 mm at the narrowest point (at the tip of the cone), the gap was 0.923 mm at the edge of the cone. The measurement methodology was set appropriately, as the maximum size of the solid particles in the chocolate masses is 25 μm [14,15]. The measurements of the chocolate masses were performed at five different temperatures of 36, 38, 40, 42, and 44 °C. Each measurement was repeated three times at an increasing shear strain rate (from 1 to 500 s^−1^) and three times at a decreasing shear strain rate (from 500 to 1 s^−1^) each time for 60 s. The average values of the shear stress were then used for the mathematical modelling.

### 2.3. Hysteresis Loop Test

Thixotropic behaviour of five different chocolate masses was characterize by hysteresis loop test [16]: Shear rate was increased logarithmically from 1 to 500 s^−1^ in 1 min, held constant at 500 s^−1^ in 10 s, and decreased logarithmically from 500 to 1 s^−1^ in 1 min. The flow measurements were performed by RST CC rotary rheometer (AMATEK Brookfield, Middleboro, MA, USA) with cone-plate geometry equipped with an RCT-50-2 spindle (diameter of the cone was 50 mm with an angle of 2°) at the temperatures of 36, 38, 40, 42, and 44 °C. The hysteresis area was calculated with the use the corporate software, Rheo3000 (AMATEK Brookfield, Middleboro, MA, USA).

### 2.4. Mathematical Modelling

The dependence of the shear stress on the shear strain rate was described by three non-Newtonian (plastic) mathematical models. The first model was the Bingham rheological model [17]:(1)$$\tau =\tau_{0}+\eta_{B}\dot{\gamma }$$

The second model was the Herschel–Bulkley rheological model [18]:(2)$$\tau =\tau_{0}+K{\dot{\gamma }}^{n}$$

And the third model was the Casson rheological model [19]:(3)$$\sqrt{\tau }=\sqrt{\tau_{0}}+\sqrt{\eta_{C}\dot{\gamma }}$$
which can also be expressed as:(4)$$\tau =\tau_{0}+2\sqrt{\tau_{0}\eta_{C}\dot{\gamma }}+\eta_{C}\dot{\gamma }$$
where τ is the shear stress [Pa], $\tau_{0}$ is the yield stress [Pa], $\eta_{B}$ is the Bingham plastic viscosity [Pa·s], $\dot{\gamma }$ is the shear strain rate [s^−1^], *K* is the consistency factor [Pa·s^n^], *n* is the flow index [–], and $\eta_{C}$ is the Casson plastic viscosity [Pa·s].

### 2.5. Statistical Analysis

The experimental data were processed by MATLAB R2018b (MathWorks, Natick, MA, USA) and Statistica 12 (StatSoft, Tulsa, OK, USA). To determine a statistically significant difference between the measured values of the shear stress and dynamic viscosity at the selected temperatures of the individual chocolate masses, an ANOVA (analysis of variance) test was performed followed by multiple comparisons using Tukey’s HSD (honestly significant difference) test (at the significance level of *p* < 0.05).

The normality and homogeneity of the data were evaluated according to a critique of the regression triplet data, which examines the quality of the data for the proposed model. The evaluation was performed by means of a graphical (using rankit Q-Q graphs of residues and index graphs) and numerical statistical analysis of the residues.

The degree of accuracy and suitability of the rheological models were evaluated using the coefficient of determination R^2^ and the sum of squared estimate of errors (SSE). The SSE is the sum of squares of the deviations of the data values from the predicted values (typically predicted from a Least Squares Analysis) using the general formula:(5)$$SSE=\overset{n}{\underset{i=1}{\sum }}{\left[y_{i}-f\left(x_{i}\right)\right]}^{2}$$
where $y_{i}$ is the *i*th value of the variable to be predicted, $x_{i}$ is the *i*th value of the explanatory variable, and $f\left(x_{i}\right)$ is the predicted value of $y_{i}$.

## 3. Results and Discussion

### 3.1. Rheological Properties

In the first phase, the mathematical description of the shear stress and shear strain rate dependence was carried out. For this purpose, the Bingham (B), Herschel–Bulkley (H–B), and Casson (C) model were used. The results of the mathematical modelling are given in Table 3 and Table 4.

From the coefficients of rheological models, it is evident that with an increasing shear deformation rate, the chocolate masses reach higher values of plastic viscosity ($\eta_{B}$, $\eta_{C}$), resp. coefficient of consistency (*K*) than with the decreasing shear strain rate. With the increasing temperature for the individual samples, these values decrease [20].

All three rheological models used confirmed the plasticity of the chocolate masses, respectively, the samples showed a certain (non-negligible) flow limit [17,20], with the highest values of this limit being reached by DC and the lowest by RC. This may be due to the different proportions of non-fat and milk components [3,21].

The flow index (*n*) in the Herschel–Bulkley model confirmed the shear thinning (with the yield stress value) behaviour (*n* < 1) of the chocolate masses, which were not significantly affected by the temperature. With a decreasing shear strain rate, the shear thinning behaviour, especially in the MC, WC, and RC samples, was insignificant.

As the high values of the coefficient of determination R^2^ and low values of the sum of the squared estimate of errors SSE (Equation (5)) prove, the used rheological models are very suitable. The Herschel–Bulkley rheological model [22] seems to be the most suitable (based on the R^2^ and SSE values) for the chocolate masses, although the Casson model is also used for chocolate masses in the literature [19,20,21,23]. The values of the statistical indicators R^2^ and SSE are given in Table 5.

### 3.2. Hysteresis Area

Subsequently with rheological measurements, the hysteresis areas were calculated [16]. The highest values of the hysteresis area were reached by the MC and WC masses at a temperature of 36 °C; on the contrary, the lowest values were reached in the RC sample. All the area values are detailed in Table 6.

The magnitude of the hysteresis area was significantly affected by the temperature; with an increasing temperature, the value of area decreased and, thus, the degree of thixotropic behaviour of the chocolate masses. With an increasing temperature, the chocolate masses, with the exception of the RC masses, took less time to reaggregate their original structure [24]. An increase in the hysteresis area at 44 °C was observed for the EDC and DC masses, due to the different structure of these high cocoa and low-fat masses [25].

### 3.3. Pipe Flow Properties

The compilation of the flow curves using the rheological models not only serves to determine the flow behaviour of liquids, but the values of the coefficients can be further used to model the physical-mechanical states of liquids, e.g., during their flow [26,27]. It is possible to calculate using the coefficients of the rheological models, for example, the mean and maximum flow velocity, volume and mass flow, coefficients of friction at the wall-liquid interface, and Reynolds number, which can be used to distinguish laminar, transient, and turbulent flow, etc. [28]. Modelling the velocity profiles of a liquid after stabilisation of both two-dimensional and three-dimensional flow seems to be the most beneficial for technical practice. For Newtonian fluids, for which the viscosity does not change with an increasing shear strain rate, such modelling is quite common and simple [29,30,31,32], but for non-Newtonian models, it is more difficult [33,34].

During tempering and other methods of processing, chocolate masses are often transported by pipelines, where the velocity profiles can be modelled using modified relationships for non-Newtonian fluids.

For Newtonian fluids, the flow velocity at any point in the pipeline can be calculated according to:(6)$$v=\frac{∆p}{4\;\eta \;L\;\left(R^{2}-r^{2}\right)}$$
where *v* is the flow velocity, ∆p is the pressure drop in the pipe, η is the dynamic viscosity, *L* is the length of the pipe, *R* is the radius of the pipe, and *r* is the distance from the pipe axis.

By modifying Equation (6), it is possible to obtain the following Equations (7)–(9), which are not widely used in the FoodTech sciences, but rather in construction materials [35], medicine and pharmaceuticals [36], the chemical industry [37], and specific polymeric materials [38]. However, as demonstrated above, these relationships are also applicable for the purpose of modelling the flow behaviour of foods and food raw materials.

To calculate the flow velocity of Bingham fluids, it is necessary to adjust the relationship and introduce the appropriate coefficients of the model [39]:(7)$$v=\frac{∆p\;R^{2}}{4\;L\;\eta_{B}}\left[1-{\left(\frac{r}{R}\right)}^{2}\right]-\frac{r\;\tau_{0}}{\eta_{B}}\left(1-\frac{r}{R}\right)$$
where $\eta_{B}$ is the Bingham plastic viscosity and $\tau_{0}$ is the yield stress (using Bingham rheological model).

To calculate the flow velocity of liquids described by the Herschel–Bulkley rheological model, it is necessary to adjust the relationship and introduce the appropriate coefficients of the model [40]:(8)$$v=\frac{n}{1+n}{\left(\frac{∆p}{L\;K}\right)}^{\frac{1}{n}}\left[{\left(R-R_{*}\right)}^{\frac{n+1}{n}}-{\left(r-R_{*}\right)}^{\frac{n+1}{n}}\right]$$
where the plug radius $R_{*}=\frac{2\;L\;\tau_{0}}{\Delta p}$, *n* is the flow index, *K* is the consistency coefficient, and $\tau_{0}$ is the yield stress (using Herschel–Bulkley rheological model).

To calculate the flow velocity of liquids described by Casson’s rheological model, it is necessary to adjust the relationship and introduce the appropriate coefficients of the model [41]:(9)$$v=\frac{R\;\tau_{w}}{2\;\eta_{C}{}^{2}}\left\{\left[1-{\left(\frac{r}{R}\right)}^{2}\right]-\frac{8}{3}{\left(\frac{\tau_{0}}{\tau_{w}}\right)}^{\frac{1}{2}}\left[1-{\left(\frac{r}{R}\right)}^{\frac{3}{2}}\right]+2\frac{\tau_{0}}{\tau_{w}}\left(1-\frac{r}{R}\right)\right\}$$
where the wall stress $\tau_{w}=\frac{\Delta p\;R}{2\;L}$, $\eta_{C}$ is the Casson plastic viscosity, and $\tau_{0}$ is the yield stress (using Bingham rheological model).

Using coefficients of the rheological models (Equations (1)–(4)) and formulas above (Equations (7)–(9)), it is possible to model the 2D velocity profiles. If we consider the standard diameter of the pipe, *D* = 50 mm; the length of the pipe, *L* = 1 m; and the pressure drop, Δ*p* = 10 kPa, the different flow behaviours of chocolate masses can be seen from Figure 1, Figure 2 and Figure 3. The x-axis on Figure 1, Figure 2, Figure 3, Figure 4 and Figure 5 shows r/R, which is the dimensionless ratio of the distance from the pipeline axis to the fixed radius of the pipeline (*D*/2). It is certain that in the axis of the pipeline, the highest flow velocity is always the highest for real liquids.

Figure 1 models the EDC sample flow velocity using all three rheology models (1)–(3) and Equations (6)–(8) used above at temperatures of 36, 40, and 44 °C.

Figure 2 models the flow velocity of the MC sample using the Herschel–Bulkley model (Equation (2)) coefficients and Equation (8). The figure shows the temperature dependence of this chocolate mass, when the mass flows the fastest and vice versa at the highest temperature (44 °C) [20,21,42].

Figure 3 illustrates the different flow velocities of all five chocolate masses at a comparative temperature of 40 °C. In this case, Casson model (Equations (3) and (9)) were used.

In the above figures, the non-Newtonian behaviour of the chocolate masses can be seen, where the flow front is typically flattened, as reported by, for example [40,43]. This can, of course, cause complications in real technical practice—e.g., turbulence [44], even when using the current trends in the food industry, such as 3D food printing, where problems may occur with the supply of the liquid or semi-liquid raw material to the nozzle, clogging of the nozzle itself, etc. [45,46].

For a realistic idea and comparison of the flow properties of the samples, it is appropriate to supplement the modelling with three-dimensional velocity profiles of flowing chocolate masses. Figure 4 and Figure 5 were created by rotating Equations (7)–(9) around the pipeline axis using Matlab. Figure 4 shows the velocity profiles of all five chocolate masses used at a reference temperature of 38 °C and using the Casson model (Equations (3), (4) and (9)). Under the same conditions (temperature, length and diameter of the pipeline, and pressure drop), the chocolate masses show different flow velocities and partly also the shapes of the 3D velocity profile. In Figure 5, the DC velocity profiles at temperatures from 36 °C to 44 °C are modelled as an example using the Casson model (Equations (3), (4) and (9)). It can be seen from the figure that with an increasing temperature, the DC flow velocity also increased and the velocity profile approached a Newtonian liquid in shape.

If the flow front in Figure 4 and Figure 5 is flattened, it is a non-Newtonian behaviour, or transient or turbulent [47] flow may also occur. This would then be interesting for further research, such as the calculation of the Reynolds number. However, this is also possible using the coefficients of rheological models, see [28].

## 4. Conclusions

Based on the obtained experimental and calculated results and their discussion, it can be stated that both hypotheses were confirmed.

With an increasing temperature, the plastic viscosity or the consistency (depend on rheological model) of the chocolate masses also resulted in an increase in the flow velocity of the chocolate masses in the pipe. The size of the hysteresis area decreased with an increasing temperature and, thus, the degree of the thixotropic behaviour of the chocolate masses. Thus, with an increasing temperature, the chocolate masses, with the exception of the RC masses, needed less time to reaggregate their structure.

The Bingham, Herschel–Bulkley and Casson rheological models can be successfully used to construct the flow curves of the chocolate masses. According to statistical preachers (R^2^, SSE), the Herschel–Bulkley and Casson rheological models appear to be the most suitable. The coefficients of the rheological models can be used for further modelling. For example, the 2D and 3D velocity profiles that are very well applicable in practice. The technical equations used to model the flow velocity of liquids at any point between the axis and the pipeline wall can be successfully used to model other foods and food raw materials with plastic or pseudoplastic rheological behaviour.

The results of modelling the flow behaviour of the food and food raw materials can be used in conventional technical practice in the design of technological equipment structures and in current trends in the food industry, such as 3D food printing.

## Figures and Tables

**Figure 1 foods-10-02519-f001:**
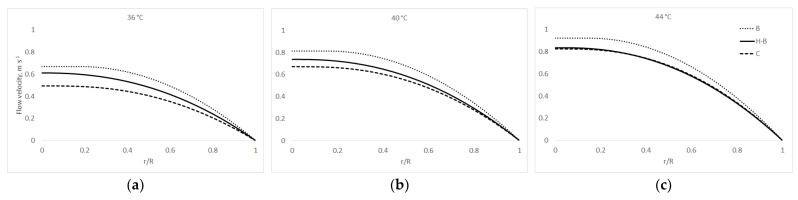
EDC velocity profiles: at (**a**) 36 °C; at (**b**) 40 °C; and at (**c**) 44 °C.

**Figure 2 foods-10-02519-f002:**
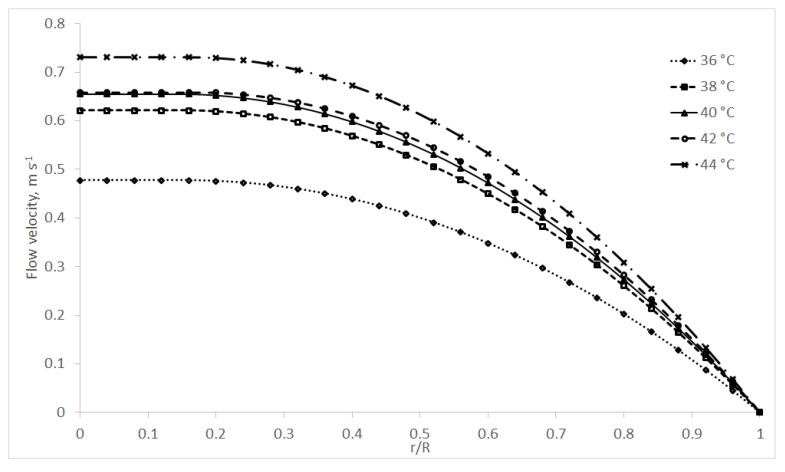
MC velocity profiles at temperatures of 36–44 °C using the H–B model.

**Figure 3 foods-10-02519-f003:**
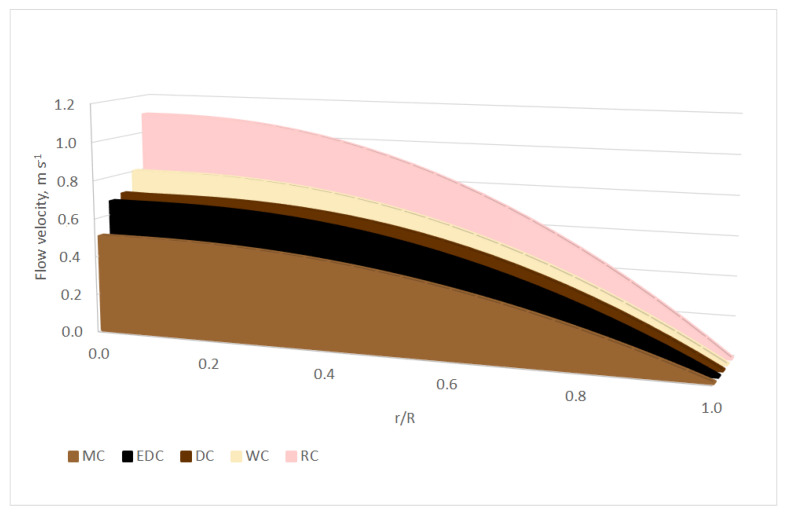
2D velocity profiles of the chocolate masses at 40 °C using the Casson model.

**Figure 4 foods-10-02519-f004:**
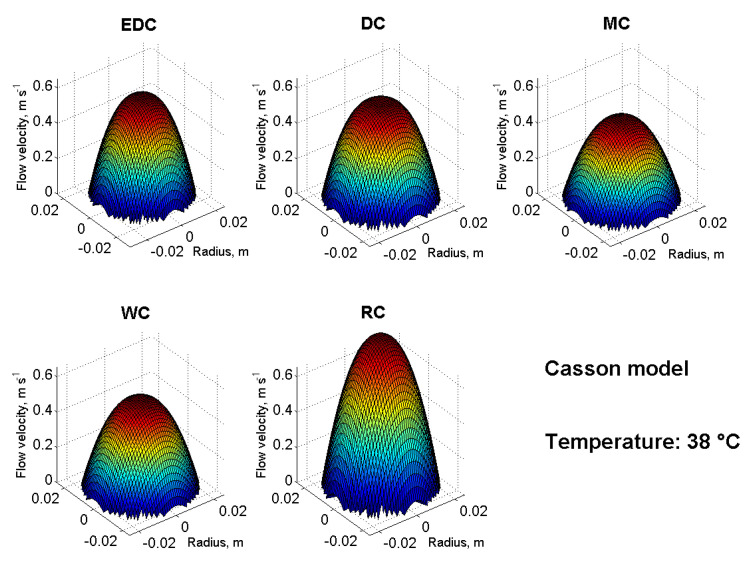
3D velocity profiles of the chocolate masses at 38 °C using the Casson model.

**Figure 5 foods-10-02519-f005:**
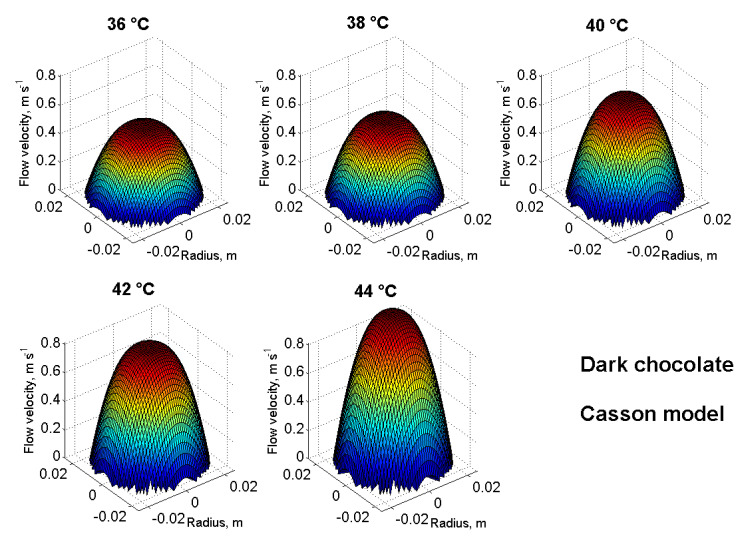
3D velocity profiles at 36–44 °C of DC using the Casson model.

**Table 1 foods-10-02519-t001:** Nutritional data of the chocolate masses.

Chocolate Mass	Nutritional Information Per 100 g					Composition
	Energy Value	Fats	Carbohydrates	Proteins	Salt	
		Of Which Saturated Fatty Acids	Of Which Sugars			
EDC	2576.0 kJ	53.0 g	14.6 g	15.0 g	0.2 g	100% cocoa components
		20.0 g	1.0 g			
DC	2264.6 kJ	35.0 g	46.9 g	5.7 g	<0.005 g	55.5% cocoa components, sugar, cocoa butter (35.0%), emulsifier: soy lecithin (E322), aroma: natural vanilla
		21.6 g	44.0 g			
MC	2297.2 kJ	33.3 g	55.6 g	5.8 g	0.206 g	36.5% cocoa components, sugar, cocoa butter (29.0%), whole milk powder, lactose, whey powder, emulsifier: soy lecithin (E322), aroma: natural vanilla
		20.4 g	54.7 g			
WC	2360.5 kJ	35.1 g	56.1 g	6.4 g	0.234 g	28% cocoa components, sugar, cocoa butter (28%), whole milk powder, emulsifier: soy lecithin (E322), aroma: natural vanilla
		21.3 g	56.1 g			
RC	2356.0 kJ	35.9 g	49.6 g	9.3 g	0.270 g	47.3% cocoa components, sugar, cocoa butter, whole milk powder, emulsifier: soy lecithin (E322), acidity regulator: citric acid, aroma: natural vanilla
		21.5 g	48.5 g			

EDC—extra dark chocolate, DC—dark chocolate, MC—milk chocolate, WC—white chocolate, RB—ruby chocolate.

**Table 2 foods-10-02519-t002:** Specific tempering temperatures of chocolate masses.

	Melting Temperature, °C	Cooling Temperature, °C	Working Temperature, °C
EDC	55	28.5–29.5	29.5–30.5
DC	45–50	28–29	31–32
MC	40–45	27–28	30–31
WC	40–42	25–26	28–29
RC	45	27	28.5–29.5

**Table 3 foods-10-02519-t003:** Coefficients of the rheological models with an increasing shear strain rate (*n* = 3).

1→500 s^−1^	T, °C	Bingham Model				Herschel–Bulkley Model					Casson Model			
		τ_0_ (Pa)	η_B_ (Pa·s)	SSE	R^2^	τ_0_ (Pa)	K (Pa·s^n^)	n (−)	SSE	R^2^	τ_0_ (Pa)	η_C_ (Pa·s)	SSE	R^2^
EDC	36	22.27	1.572	19,340	0.9914	0.870	4.688	0.8185	1415	0.9994	5.865	1.276	6045	0.9973
	38	23.21	1.456	17,460	0.9909	3.002	4.415	0.8158	1506	0.9992	6.704	1.168	5314	0.9972
	40	19.79	1.360	10,640	0.9937	4.306	3.498	0.8429	825	0.9995	5.355	1.115	2844	0.9983
	42	18.22	1.291	8090	0.9946	4.684	3.122	0.8531	487	0.9997	4.886	1.068	1888	0.9987
	44	18.62	1.225	7402	0.9946	5.820	2.951	0.8537	587	0.9996	5.297	1.006	1768	0.9987
DC	36	32.92	1.462	5272	0.9973	22.28	2.788	0.8925	319	0.9998	9.959	1.133	1012	0.9995
	38	31.68	1.389	5744	0.9967	20.48	2.814	0.8825	322	0.9998	10.050	1.075	886	0.9995
	40	27.73	1.257	2731	0.9981	20.29	2.150	0.9105	225	0.9998	8.695	0.989	1007	0.9993
	42	22.68	1.035	1700	0.9982	17.96	1.581	0.9293	626	0.9994	7.543	0.827	1705	0.9982
	44	30.08	1.101	3728	0.9966	20.86	2.284	0.8785	144	0.9999	11.891	0.835	300	0.9997
MC	36	26.29	2.018	4058	0.9989	18.29	2.927	0.9380	1001	0.9997	3.724	1.718	2505	0.9993
	38	21.51	1.702	835	0.9997	18.48	2.026	0.9709	373	0.9999	2.945	1.478	2644	0.9990
	40	21.26	1.612	1116	0.9995	17.40	2.032	0.9614	371	0.9998	3.277	1.387	2225	0.9990
	42	24.84	1.486	1022	0.9995	20.59	1.954	0.9544	148	0.9999	5.199	1.233	2054	0.9990
	44	22.07	1.429	979	0.9995	19.80	1.670	0.9740	718	0.9996	4.137	1.210	3246	0.9982
WC	36	22.95	1.802	4749	0.9984	13.17	2.961	0.9173	338	0.9999	3.802	1.528	930	0.9997
	38	21.04	1.606	3125	0.9987	13.29	2.509	0.9256	314	0.9999	3.743	1.364	915	0.9996
	40	18.43	1.264	1541	0.9990	13.88	1.776	0.9433	450	0.9997	3.937	1.072	1266	0.9991
	42	24.24	1.472	6122	0.9969	12.67	2.936	0.8851	310	0.9998	6.033	1.199	736	0.9996
	44	24.27	1.393	3762	0.9979	15.84	2.410	0.9087	562	0.9997	6.141	1.135	951	0.9995
RB	36	16.07	1.385	1543	0.9991	11.05	1.951	0.9429	323	0.9998	2.709	1.205	853	0.9995
	38	14.40	1.258	564	0.9996	11.32	1.593	0.9606	96	0.9999	2.410	1.104	703	0.9995
	40	11.68	1.123	368	0.9997	9.80	1.323	0.9726	190	0.9998	1.848	1.002	784	0.9993
	42	14.21	1.071	312	0.9997	12.07	1.302	0.9675	83	0.9999	2.944	0.929	808	0.9992
	44	16.21	1.020	237	0.9997	14.98	1.149	0.9801	159	0.9998	3.857	0.868	1579	0.9983

**Table 4 foods-10-02519-t004:** Coefficients of the rheological models at a decreasing shear strain rate (*n* = 3).

500→1 s^−1^	T, °C	Bingham Model				Herschel–Bulkley Model					Casson Model			
		τ_0_ (Pa)	η_B_ (Pa·s)	SSE	R^2^	τ_0_ (Pa)	K (Pa·s^n^)	n (−)	SSE	R^2^	τ_0_ (Pa)	η_C_ (Pa·s)	SSE	R^2^
EDC	36	17.98	1.491	15,120	0.9925	0	4.071	0.8330	1310	0.9994	4.477	1.239	5004	0.9975
	38	19.46	1.406	8048	0.9955	8.258	2.827	0.8837	2416	0.9986	4.428	1.175	3709	0.9979
	40	18.12	1.319	6449	0.9959	8.750	2.480	0.8949	2410	0.9985	4.110	1.110	3481	0.9978
	42	13.43	1.279	5338	0.9964	3.092	2.594	0.8823	623	0.9996	2.882	1.102	1660	0.9989
	44	15.84	1.201	4332	0.9967	9.894	1.896	0.9238	2592	0.9980	3.344	1.029	3594	0.9972
DC	36	21.94	1.462	2087	0.9989	15.46	2.210	0.9312	128	0.9999	4.563	1.224	706	0.9996
	38	21.26	1.380	2599	0.9985	14.18	2.213	0.9213	279	0.9998	4.801	1.149	761	0.9996
	40	15.86	1.276	4457	0.9970	7.97	2.231	0.9069	1776	0.9988	3.541	1.086	1334	0.9991
	42	20.85	1.016	1342	0.9986	18.84	1.232	0.9680	1134	0.9988	6.225	0.830	3254	0.9965
	44	18.38	1.098	1160	0.9989	13.54	1.655	0.9316	63	0.9999	4.938	0.913	503	0.9995
MC	36	19.30	1.854	2625	0.9992	16.41	2.161	0.9744	2213	0.9993	2.077	1.644	3570	0.9988
	38	15.28	1.665	219	0.9999	15.55	1.638	1.0030	216	0.9999	1.309	1.516	2461	0.9990
	40	17.22	1.583	278	0.9999	17.28	1.577	1.0010	278	0.9999	1.814	1.419	3277	0.9985
	42	18.93	1.461	1528	0.9992	22.00	1.170	1.0370	997	0.9995	2.134	1.303	7279	0.9962
	44	11.85	1.436	2491	0.9987	8.95	1.748	0.9672	2086	0.9989	1.415	1.296	2011	0.9989
WC	36	17.60	1.658	1943	0.9992	16.17	1.806	0.9857	1840	0.9993	1.953	1.477	3757	0.9985
	38	11.47	1.575	2314	0.9990	6.76	2.095	0.9524	1274	0.9994	1.276	1.427	956	0.9996
	40	12.44	1.231	178	0.9999	11.13	1.367	0.9824	91	0.9999	1.737	1.102	854	0.9994
	42	14.97	1.440	408	0.9998	12.63	1.688	0.9735	134	0.9999	1.950	1.281	1107	0.9994
	44	10.92	1.385	2312	0.9987	6.88	1.832	0.9533	1542	0.9991	1.418	1.248	1280	0.9993
RB	36	12.630	1.373	1029	0.9994	12.301	1.407	0.9959	1023	0.9994	1.233	1.254	2768	0.9984
	38	9.364	1.255	157	0.9999	9.067	1.285	0.9960	152	0.9999	0.856	1.162	895	0.9994
	40	9.158	1.108	61	0.9999	8.859	1.138	0.9955	56	0.9999	1.035	1.018	755	0.9993
	42	9.287	1.067	77	0.9999	9.355	1.060	1.0010	77	0.9999	1.109	0.979	876	0.9991
	44	9.391	1.017	35	0.9999	9.561	1.001	1.0030	33	0.9999	1.231	0.929	862	0.9991

**Table 5 foods-10-02519-t005:** Values of the statistical indicators of the rheological models (*n* = 50).

Model	Coefficient of Determination R^2^					Sum of Squared Estimate of Errors SSE				
	Mean	Median	MIN	MAX	SD	Mean	Median	MIN	MAX	SD
Bingham	0.9980	0.9989	0.9909	0.9999	0.0023	3581	2200	35	19,340	4318
Herschel–Bulkley	0.9996	0.9998	0.9980	0.9999	0.0004	733	372	33	2592	729
Casson	0.9988	0.9991	0.9962	0.9997	0.0008	2034	1457	300	7279	1547

MIN—minimal value, MAX—maximal value, SD—standard deviation, H–B—Herschel–Bulkley model.

**Table 6 foods-10-02519-t006:** Hysteresis values of the chocolate masses (*n* = 3).

Temperature, °C	EDC, (Pa·s^−1^)	DC, (Pa·s^−1^)	MC, (Pa·s^−1^)	WC, (Pa·s^−1^)	RC, (Pa·s^−1^)
36	12 814	7521	25 787	21 694	3625
38	9531	6771	7953	8817	3147
40	6554	3705	6156	7657	3266
42	3901	3446	6892	9063	3238
44	4924	6318	4501	8363	3952

## Data Availability

The data presented in this study are available on request from the corresponding author.

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
