# Peer review of "Rheological and Pipe Flow Properties of Chocolate Masses at Different Temperatures"

_foods, 2021, doi:10.3390/foods10112519_

Round 1
Reviewer 1 Report
foods-1392173-peer-review-v1
The authors carry out a simple measurement using rheology where they measured the shear stress as a function of the shear strain rate. Although, later on they talked about “shear deformation rate”, so it is not clear what they measured as it was not well articulated in materials and methods. Three different well known models were used to fit the rheological data to compute the viscosity and other parameters according to each model used. R2 and SSE were used to evaluate the fit of the data to the model and the repetitions. The measurements were performed at six different temperatures and five different commercial chocolates were used. The description of the 5 commercial chocolates is poor.
None of the experiments carried out here are new as many researchers have done this kind of measurement through the years. What is new is that the authors tried to extend the information obtained to be used with equations that considered the flow of liquid in a pipe line. They used reference 27 as the basic for what they did, but the whole manuscript is neither well written or well explained. Also, not all figures are well presented. They need to clearly indicate the variables in the X and Y axis as well as from where those figures were obtained.
The hypothesis are badly presented. Those are not hypothesis, they are statements and in the first case, while writing the hypothesis, the authors were trying to explain it. The whole purpose of a hypothesis is to make a statement and then present the data and a discussion to either ratified the hypothesis or to negate it.
The correction to a Newtonian equation has to be well articulated. The authors cite reference 38, which is a proceeding to a conference rather than citing a well stablish book. It is not clear why equation 6 is only valid when the Bingham model is used to obtain τo. The authors then send the reader to look at reference 39, when they can add one or two sentences to explain what they are doing. It is not clear why they say “flow rate” for equation 6 and they use the same symbol as in equation 5 which was called “flow velocity”.
When the authors talk about 2D modeling, it seems that it is a cross section of the tube that they are modeling as r/R indicates. Nowhere did the authors explained that.
Not clear what they do when they generated the plots for Figure 4 and 5 as the text does not match the figure (the text mentioned one temperature but the figures showed many different temperatures). They mentioned that they use equation 8 and the Casson model, perhaps they should elaborate more as these plots are what distinguish this work from others. And why is there no discussion about the results from Figure 4 and 5?
The conclusions are not well articulated. They over reached as half of what is written was not proved by the authors with the experiments carry out here.
Other minor comments
Line 36: “limit values”, limit values of what? The sentence needs to be better written.
The authors are stating minimum percentage of cocoa butter and other ingredients but they are failing to say for which country this applies, as we all know that each country has their own legislations.
Line 41-42. white chocolate is described as a “delicacy” but the end of the sentence calls for “other ingredients”, which will not indicate that it is a “Delicacy” as the reader does nto know what those ingredients are. Fix the sentence
Line 44-46 is not well written.
In line 47 the authors mention a fourth chocolate type , “Ruby” but later on , on line 52 they said that the ruby is not yet recognize as a different category. Fix the sentence in line 47 as it does not agree with what it is being said in 52
Line 61 should also cite references like Beckett or Afoakwa.
Lines 68-72 are not well described. “beginning of chocolate flow”? “some tension”? “Flow limit”? Make the reading clear for the reader.
Material and Methods
The introduction discusses that chocolates should contain cocoa butter, fat-free cocoa solids, possible milk fat, milk solids and sweeteners or “other ingredients”. The authors failed to explain what “cocoa solids” are, as they claim in line 93 that this is the only ingredients in the EDC chocolate ( how much cocoa butter is in the EDC? How much is fat free cocoa solids?). Be consistent all along the paper.
It seems that all chocolates studied here contain different amounts of fats, coming from cocoa butter or milk fat. As we all know, fat is responsible for the flow of the chocolate mass. Table 1 is not clear: it says ‘Fats’ and below it “of which saturated fatty acids”. A comment about the fats should be stated if the authors want to discuss that and matched it with why for example the ruby profile of velocity seems to be higher that others. Otherwise, why bother with the fat content? After all, the authors are using commercial chocolate available on the market to anyone
Author Response
Dear reviewer,
thank you very much for your very helpful comments, which in the vast majority of cases we have incorporated into the text, which will greatly enrich it.
The response to the individual comments is attached in a separate document.

Reviewer 2 Report
The assessed work concerns an important issue and may be of great practical importance. After reading it, he has a few minor comments and questions.
How do the authors justify the selection of the temperature range in which the tests were carried out?
How were the nutritional values determined? They were chemically marked - according to what procedures ?. calculated? -
I suggest extending the assessment of the fit of the models with reduced chi-squared statistic and provide the formula for which the sum of squared estimate of errors (SSE) was calculated.
Author Response

(The authors gave the same response as above.)

Reviewer 3 Report
Interesting and work having the aspect of technological use. New products used in the raw chocolate - pink chocolate.
lined 62-64 "The rheological and textural properties of the chocolate are affected by its specific composition, especially the content of the fats, sugars, milk, other milk components and surfatants. The fat content of chocolates is usually in the range of 25 –35% [3]. the authors mention the role of fat on the rheological properties of chocolate masses. I think that in order to increase the scientific value of the work, it is worth adding a few sentences about the type of cocoa butter (A, b, c) and the influence of each of them on the crystallization of chocolate mass, the effect on the final equilibrium ... that is, from a scientific point of view, mention tempering.
2.1. Samples of Chocolate Masses for easier interpretation of the results, they should be selected - tested chocolate masses from the same manufacturer. The processes used in the production and shaping of chocolate, which are often a secret of the manufacturer, can be of great importance for the rheological properties.
tab 1 - is it the full composition of the chocolate masses or are there any emulsifying aid (lecithin)?
2.3. Hysteresis Loop Test please discuss the execution in more detail, this is the key of discussing the results. the same citation [16] - yes, it is justified, but by very popular type designations. water, protein, etc. content. Even the ambient temperature level plays a key role in determining the thixotropy of the mass.
table 5 I do not understand the value of the hysteresis loop at the temperature level of 36 degrees Celsius if it is 12 thousand, then without a dot or a decimal point, and if it is 12 - it is very confusing and misleading to present tenths-hundredths-thousandths. Please correct!
3.2. Hysteresis area I think that in this chapter it is necessary to add what technological, and in principle "life of the product" storage on the shelf, is important for hysteresis and its size
graphs 3 and 4 should have the same maximum value describing the axis "Flow .."
conclusions Result conclusions summarizing the research theses. it is very good to indicate the possibility of using the results of work in the design of technological processes in the confectionery industry and to predict changes in chocolate mass under various temperature conditions. Conclusions should be written as continuous text, without dotted paragraphs
Author Response

(The authors gave the same response as above.)

Round 2
Reviewer 1 Report
The creating of the 3D plots might help industry in their design of pipelines as the equation can easily be modified for other pipe diameters.
The write up needs minor corrections and an explanation for the tempering of the chocolates analyzed.
Incomplete sentence: "In this research, a description of the behavior of chocolate masses that do not contain cocoa butter substitutes".
The authors state that they have 3 hypothesis but they only listed 2. Fix the text
Not sure why in Table 1 they have two set of values for some chocolates. Which one did they use for the analysis?
Table 2: where did the authors get those values for tempering? No literature is shown and there was no confirmation that the chocolate was in fact tempered ( no DSC or XRD results discussed)
It is not clear why they carry out a “Tempering” of the commercial chocolate bought when the rheology was carried out at temperatures higher than 36°C, hence destroying the tempering. Needs explanation.
This sentence should be more specific : Technical equations for determining the flow velocity rate of liquids at any
point in the pipeline can be successfully used for food and food raw materials.
Author Response
Dear reviewer,
thank you very much again for your very helpful comments, which will greatly enrich our paper.
The response to the individual comments is attached in a separate document.
Best regards,
Vojtech Kumbar
